# The functional generalization of the Boltzmann-Vlasov equation and its Gauge-like symmetry

Giorgio Torrieri

*Universidade Estadual de Campinas - Instituto de Fisica "Gleb Wataghin"*

*Rua Sérgio Buarque de Holanda, 777*

*CEP 13083-859 - Campinas SP*

*torrieri@ifi.unicamp.br*

We argue that one can model deviations from the ensemble average in non-equilibrium statistical mechanics by promoting the Boltzmann equation to an equation in terms of *functionals* , representing possible candidates for phase space distributions inferred from a finite observed number of degrees of freedom.

We find that, provided the collision term and the Vlasov drift term are both included, a gauge-like redundancy arises which does not go away even if the functional is narrow. We argue that this effect is linked to the gauge-like symmetry found in relativistic hydrodynamics [1] and that it could be part of the explanation for the apparent fluid-like behavior in small systems in hadronic collisions and other strongly-coupled small systems[2].

When causality and Lorentz invariance are omitted this problem can be look at via random matrix theory show, and we show that in such a case thermalization happens much more quickly than the Boltzmann equation would infer. We also sketch an algorithm to study this problem numerically

# I. INTRODUCTION

The problem of apparent hydrodynamic behavior of small systems [2] is one of the most, if not the most important conceptual problem thrown at us by heavy ion collisions. Experimental data [3, 4] seems to suggest that "collectivity" (precisely defined as the number of particles present in correlations relevant for anisotropic flow) is remarkably insensitive to the size of the system produced in hadronic collisions, down to proton-proton and $\gamma - nucleus$ collisions with 20 final state particles.

Most of the theoretical response to this has been centered around the concept of "hydrodynamic attractors"/hydronamization [5, 6], based on the idea of taking a "microscopic" theory (usually Boltzmann equation, a theory with a gravity dual or classical Yang-Mills) in a highly symmetric (lower dimensional) setup and showing that hydrodynamic behavior occurs for gradients much higher than those "naively expected". The basic issue is that the main puzzle of the onset of hydrodynamics in such Small systems is not the size of the gradients, but rather the small number of degrees of freedom [7, 8], which generate fluctuations in every cell even if the mean free path was zero [9]. Yet all indications seem to show that fluctuations are reducible only to fluctuations in initial conditions [3]. This absence of fluctuations can be thought of as another sign of "perfect" hydrodynamics [10–12], which also seems to appear beyond its naive range of validity. In this regime most microscopic theories based on large $N$ approximations (Boltzmann equation with its molecular chaos, AdS/CFT in the planar limit, classical Yang Mills theories with large occupation numbers, even Kubo formulae and Schwinger-Keldysh approaches requiring asymptotic limits for soft modes [13, 15–17]) become suspect.

Similarly, approaches based on "anomalous viscosity" and plasma instabilities [18, 19] look suspect because while on average they might reproduce a low-viscosity fluid, fluctuations around this average are expected to be much larger than hydrodynamic expectations [10]. Multi-particle correlation analysis seems to suggest,on the other hand, the absence of "dynamical" fluctuations not reducible to initial state effects, even in small systems [3].

Here one must remember that a universal hydrodynamic-like behavior in small systems has been noted in a much larger set of circumstances than the debate around heavy ion collisions usually includes: Cold atoms seem to have achieved the onset of hydrodynamics with comparatively few particles [7, 8]. It has long been known that Galaxies behave "as a

fluid", even though the assumptions related to transport are highly suspect [20]. Even in everyday physics,phenomena such as the "Brazil nut effect" [21] point to a universality of the hydrodynamic description even in systems with few particles, provided they are strongly correlated. This apparent universality is cited by mathematicians such as [22] to study the multi-particle problem in depth.

Recently it was argued [1, 24] that a way to approach this conundrum is to think in a Gibbsian rather than a Boltzmannian way [25, 26]: The latter treats the phase space as a frequentist probability density distribution, and hence is in a sense well-defined only in an infinite particle limit. The former arrives at phase space distributions via Bayesian inferencing of non-measurable microscopic quantities via macroscopically measurable coarse-grained degres of freedom (the two descriptions are proven to be equivalent for an ideal gas in a thermodynamic limit only [25, 26]).

For a strongly coupled system with a small number of degrees of freedom, since only the energy-momentum tensor and conserved current components are measurable, fluctuations bring with them a redundancy of *hydrodynamic descriptions*, each with it's flow vector and Bayesian probability that this is the "true flow vector" and any anisotropy is due to a fluctuation. If the system is strongly coupled enough for the fluctuation-dissipation theorem to apply locally, each of these descriptions is as good as the others as long as the total energy-momentum is the same. It is not surprising therefore that as fluctuations become larger the probability of a good description near the ideal hydrodynamic limit could actually grow, or at least it does not go down [1]. In a sense this picture is the inverse of that of an attractor. This description parallels the role of Gauge symmetry in the renormalizability of Quantum field theory [27]: The fact that "most fluctuations" can be accomodated by Gauge redundances lowers the degree of divergences in the ultraviolet and perhaps (in the Gribov-Zwanziger picture) also the number of degrees of freedom in the infrared [28].

For now such a picture is still abstract and qualitative. In this work, we would like to make a link to microscopic theory, via a generalization of the Boltzmann-Vlasov equation which goes in the "Gibbsian sense" outlined earlier. The basic idea is that when the number of degrees of freedom is small, the phase space distribution $f(x, p)$ will not be known but must be inferred by some kind of Bayesian reasoning. This is admittedly a very heuristic approach, and some further arguments motivating it and placing it within the more conventional transport theory have been left to the next section.

The fact that one does not know $f(x,p)$ beyond a few data points can be represented by considering a *functional* representing the probability of $f(x,p)$ being what it is. In this case, the integrals corresponding to the Boltzmann collision operator and the Vlasov potential operator will not be two copies of $f(x,p)$ but two different functions $f(x,p)$ and $f'(x,p)$, the latter integrated over.

The next section II gives further details of the shortcomings of the state-of-the-art transport approaches and why a Boltzmann equation with functionals is a possible path forward in a certain well-defined regime of validity. Section III gives some mathematical details, and shows that the limit close to equilibrium of the functional Boltzmann equation is very different from the usual Boltzmann equation, exactly because of the residual ambiguities pointed out in [1]. Section IV argues this can be shown in terms of random matrices, and Section IV B suggests an algorithm to test these ideas numerically.

## II.   TRANSPORT APPROACHES AND THEIR LIMITS

### A.   Free streaming, perfect hydrodynamics and ensemble averaging

Transport theory can be thought of as a limit of a classical $N \to \infty$ particle system, where the Hamilton Jacobi equation tends to a distribution function, $x_{i=1,N}, p_{i,N} \underset{N\to\infty}{\to} f(x,p)$ [22]. The Hamiltonian evolution of this limit also tends to be infinitely unstable, with the Boltzmann collision term cutting off this instability "at the price" of time-reversibility. More complicated correlations $f(x_1, p_1, x_2, p_2, ...)$ are also possible, arranged in the so-called BBGKY hyerarchy [29]. The ideal hydrodynamic limit is reached when the Boltzmann collision term is in local equilibrium, and the Vlasov term irrelevant (either averaged out or quenched by Debye screening).

As can be seen, ideal hydrodynamics thought in this way is a coincidence of several limits, and this can lead to "paradoxes". For instance, let us take eq 6 in the free streaming collisionless case, adding a mass for the result to have a good classical limit

$$\frac{p^\mu}{m}\partial_\mu f(x,p) = 0 \tag{1}$$

Physically, an obvious solution corresponds to the Galilean motion of particles at constant

velocity that have been released

$$f(x,p) = f\left(x_0 + \frac{p}{m}t, p\right) \tag{2}$$

However, what is counter-intuitive is that not the only solution; Consider the case where the particles are in a thermal distribution according to some field $\beta_\mu(x)$. In this case, it is trivial to check that

$$f(x,p) \sim \exp\left[-\beta_\mu p^\mu\right] \quad , \quad \partial_\mu \beta_\nu + \partial_\nu \beta_\mu = 0 \tag{3}$$

also solves Eq. 1 . In particular, an irrotational vortex $\vec{v} \sim \frac{\Gamma}{2\pi r}\hat{\theta}$ will correspond to a gas rotating forever.

   Mathematically, this is understandable: The right hand side of the transport equation vanishes for both free streaming and ideal hydrodynamic limits, and in the latter the flow vector is the Killing vector. But physically, on the surface this makes no sense! How can a gas of non-interacting particles just rotate? There is no force keeping them rotating. More generally, the condition on Eq. 3 is that of a Killing vector, in line with the idea that flow is a Killing vector of the co-moving frame in ideal hydrodynamics [30, 31]. But once again, these are non-interacting particles: Pressure gradients do not correspond to any force on neighboring volume elements since particles just propagate freely. Very clearly no system of non-interacting particles, when freely released, will start "flowing". This paradox

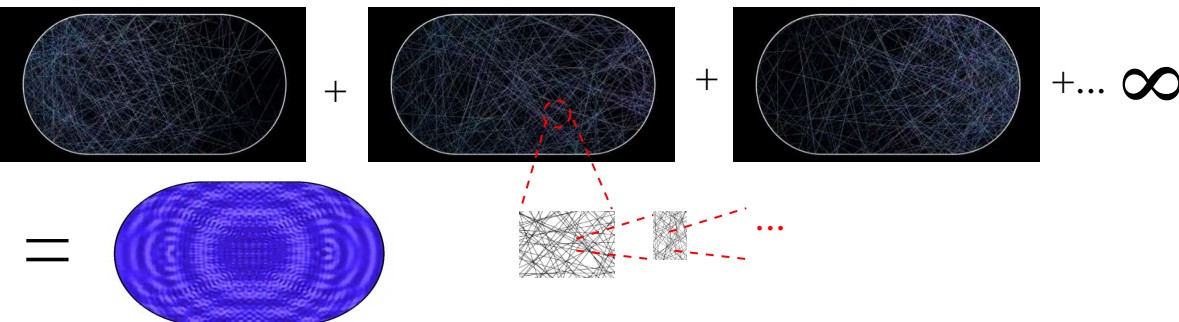

FIG. 1. A representation of how an uncountable number of physically sensible free-streaming configurations of finite trajectories can become a smooth but curved "fluid" when summed together as an ensemble average. In the ideal fluid dynamics limit, on the other hand, particle trajectories in each sub-event would follow the ensemble average

is resolved by remembering that $f(x,p)$ is defined in an ensemble average limit where the

number of particles is not just "large" but *uncountable.* Just like the limit of many straight segments is curved, once an infinite number of trajectories are summed over, the *maximum density* of trajectories can be a curve even if trajectories are straight. This means that if we divide $f(x, p)$ in any number of "physical" sub-events each with a finite number of particles, *none* of the sub-events will look like Eq. 3 , but each will look like some version of Eq. 2 . However,the number of copies of each Eq. 2 *close to its neighborhood in phase space* will have a curvature, so when this is summed over a smooth "Killing vector" emerges. This is illustrated in Fig. 1 [32] [1]. In contrast, in the ideal hydrodynamic limit,even away from the pure ensemble averaging each microscopic particle will "flow" under the action of pressure gradients, and the probability that it flows differently goes to zero in the ensemble limit. Some put this as the real definition of hydrodynamics [12]: Initial conditions and conservation laws fix the final state *for individual particles.* Note that this is what seems to emerge from multi-particle cumulant analysis of experimental data [3].

What this suggests is that, analogously to the volume in phase transitions, the ensemble average limit is *non-analytic.* Being arbitrarily close to it does not necessarily give a qualitatively similar description w.r.t. it. In other words, the transport properties of a system of finite degrees of freedom need not be close to their Boltzmann equation results even if the number of degrees of freedom is large. It also suggests that away from the non-analytic limit stochasticity due to a limited number of degrees of freedom interplays with the Knudsen length scale in a highly non-trivial way: One can regard each sub-ensemble as a frequentist "world", random scattering as the interaction "between worlds" and Vlasov evolution as a semi-classical interaction "within a world". A functional picture, where "every world" corresponds to a probabilistic ensemble of phase space functions, might be the ideal way of dealing with this picture in a consistent manner.

## B.   Transport in quantum mechanics and field theory

The arguments in the last section are fundamentally classical. Quantum mechanics comes with a further "expansion scale" $\hbar$ (in reality, $\hbar$ can be thought of as unity and the expansion

---

[1] In reality this much studied "billiard stadium" setup is a bit different, for the particles interact with the boundaries, and this provides a measure of chaos that leads to a fractal rather than continuous density profile. But the main idea behind taking a limit prevails. An "free streaming fluid" in such a stadium initialized as an ensemble limit thermalized distribution would evolve as a turbulent fluid with "fractal" scale-free flow

is around the action $S \gg \hbar$ or equivalently state occupancy). Furthermore, the expansion around $\hbar$ and $T$ do not commute. Let us review the current consensus of how transport theory fits in with quantum field theory [13, 15].

The current consensus is that in quark-gluon plasma physics the Vlasov terms are taken care of by resummation and screening. The idea is that such terms would be relevant at a "soft" scale $k \sim gT$ (where $g$ is the quantum field coupling constant), where field are classical. Thus, in a manner somewhat analogous to the argument in [22], the soft modes are taken care of by the Vlasov equation while the hard modes are put in the collision Kernel. For this to work within a quantum field theory perturbative expansion, one needs an intermediate scale, which can be the temperature or more generically the occupancy of soft states. We can then resum the soft modes

$$Vlasov \sim g^2 \left\langle A^2 \right\rangle \sim g^2 \int \frac{d^3 p}{E_p} f(E_p, T) \quad , \quad Boltzmann \sim g^2 \left\langle A^4 \right\rangle \sim f \times f \qquad (4)$$

with all interactions between them counted as a correction to the propagator. The hard mode ($k \sim T$) interactions $\sim g^2 \left\langle A^4 \right\rangle$ are then accommodated as a Boltzmann equation with distributions and collision kernels calculated via such propagators [15]. In this regime, the main effect of the fields is Debye screening and if the mean free path is well above the Debye screening length the Vlasov terms become irrelevant.

There are two issues in this description when a finite number of degrees of freedom are excited and one is well away from a thermodynamic limit: The first is that the *ultra* soft modes $k \sim g^2 T$ couple to the soft modes via Plasma instabilities and can not generally be treated perturbatively. If the boundary is fixed, for example by an asymptotic expansion around a hydrostatic state (as is done via Kubo/Schwinger-Keldysh formulae), this provides boundary conditions that render these ultra-soft modes irrelevant, but for small systems this is suspect.

The second is that while "to leading order" one can obtain a "resummed Boltzmann equation" it is not clear what the next to leading order is and how good is the convergence of this series [33]. The fact that to zeroth order in the collision term an infinite quantum thermal loop summation leads to a classical Vlasov equation [34] ilustrates how careful must we be with any such expansion: Basically the expansion in correlations, in $\hbar$ and in temperature do not commute.

What we are looking for in this work is a limit where these correlations are classical-

probabilistic rather than quantum. In this case they parametrize our ignorance of the phase space distribution rather than quantum correlations. So the question is where would this ansatz lie in corrections of the coupling constant $g$ and Planck scale $\hbar$. The different regimes

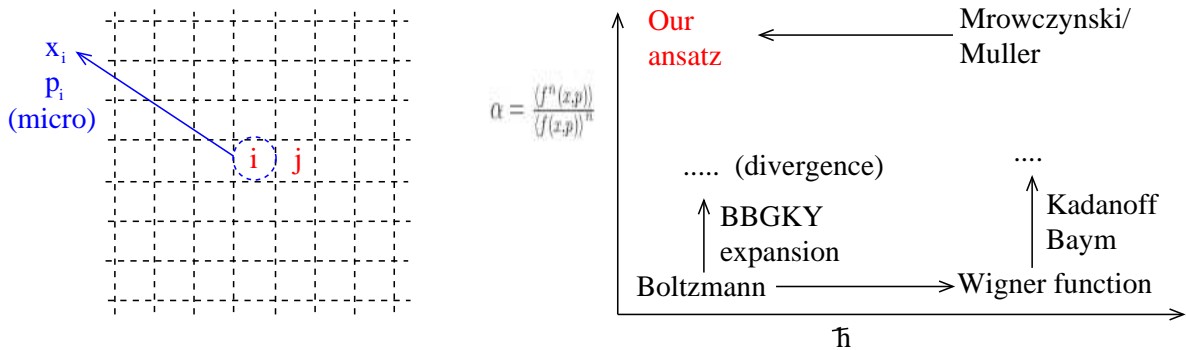

FIG. 2. The domain of validity of the ansatz proposed here, in terms of fluid cell coarse graining and microscopic variables $x_i, p_i$

of many body theory are illustrated in Fig. 2. In practice, we hope to describe a system which is

**Strongly correlated:** , so that the BBGKY hierarchy can not be used as an "expansion" but must be handled "non-perturbatively", analogously to functional methods in QFT (Gaussianity, as we shall see later, comes from renormalizeability [37–40]). This is the meaning of $\alpha = \frac{\langle f^n(x,p) \rangle}{\langle f(x,p) \rangle^n} \gg 1$. Otherwise, one can try to truncate the BBGKY hierarchy and the limit of this is the Boltzmann equation. Vlasov terms together with functionals of $f$ (including arbitrary $n$-point functions) will keep track of long-term correlations, while collision type terms will keep track of short-term ones.

**Classical-probabilistic:** in the sense that statistical independence has to hold. In other words the $CHSH$ inequality [41].

$$\langle x_i, q_j \rangle - \langle x_i, p_j \rangle + \langle x_i, q_j \rangle + \langle x_i, p_j \rangle \le 2 \tag{5}$$

must hold for any pair of conjugate observables (position,momentum, spin for fluids with polarization) from any cells $i, j$ This is required for the probabilities of any field configuration must be classical functionals rather than quantum operator averages. In this sense, $h \ll 1$ (or equivalently state occupancy $\gg 1$). Otherwise, phase space functions and functionals stop being classical objects. Note that the saturation of Eq.

5 might be done by decoherence with unseen degrees of freedom or by "Eigenstate thermalization" of a strongly coupled quantum evolution [42], something described in some detail in [24], and within effective field theory in [43]

In certain situations [44–47], when $\langle m\phi^2/\hbar \rangle \gg 1$ the spectral function of the theory is dominated by quasi-particle peaks [35] (the expansion in hard thermal loops can lead to a temperature-dependent effective mass term that in turn generates an effective thermal "force" incorporated into the Vlasov equation [36]) In this case, the assumptions above lead to a Boltzmann-Vlasov equation with a strong mean field term and features like in-medium masses and widths. However, there is no guarantee for this to happen, and such mean-field theory is not known to produce a hydrodynamic-like behavior in small systems. We therefore go in a different direction (perhaps $(\langle m\phi^2/(\hbar g^2) \rangle \gg 1, g^2 \sim 1)$, giving up the quasi-particle assumption but retaining the quasi-classical behavior and the strong correlations so that the system is represented by a classical probabilistic ensemble of fields. If these fields are both highly occupied and weakly coupled, the dynamics has analogies to that of a Color Glass condensate [48] but if the coupling is strong enough to guarantee the sort of random phase decorrelation consistent with the Eigenstate thermalization hypothesis but also occupation number is small enough that fluctuations of it are non-negligible, no ansatz is currently known. We conjecture that because of the ETH one can continue using classical probabilities rather than quantum density matrices in this regime. Since we are working in the limit of a large ensemble (perhaps with a small number of DoFs per event but "many" events) a well-defined convergent functional integral is required to construct such an ensatz. In the next section we argue that a Boltzmann-Vlasov functional with a Gaussian ansatz could be the ansatz we require, and argue that the limit to the usual Boltzmann equation could be non-analytical in the number of degrees of freedom, thereby providing a mechanism for fast thermalization of small systems

## III. MATHEMATICAL DEVELOPMENT OF A BOLTZMANN EQUATION WITH FUNCTIONALS

The idea of including fluctuations in the Boltzmann equation is not new [14], but up until now it was done in a linearized stochastic expansion.

To try to develop an exact theory let us start from the Boltzmann-Vlasov equation Eq.

$$\frac{p^\mu}{\Lambda} \frac{\partial}{\partial x^\mu} f(x,p) = C[f] - gF^\mu \frac{\partial}{\partial p^\mu} f(x,p) \qquad (6)$$

where $\Lambda$ is a generic IR momentum scale, usually associated with the particle mass, but might also be the virtuality, the Debye screening length and so on. We note that we wrote Eq. 6 in an unusual way for, other than the generic virtual scale, usually the Vlasov term is on the right-hand side. $F^\mu$ is the four-force.

This way of writing, however, is physically justified as we will show. The Vlasov term is of the same order of magnitude in the Coupling constant as the Boltzmann term, and is thought to dominate for long-range correlations where, due to Bose-enhancement, the occupation numbers of bosons are high requiring a semi-classical field. It is thought that instabilities due to thermal fluctuations and Debye screening make this term obsolete but, for "small" but highly correlated systems, there is no justification for this.

More formally, the Boltzmann equation is known to be a good approximation of a quantum field theory evolution in the "ensemble average". One way to express this is to consider that the Wigner functional can be approximated by a one particle Wigner function which in turn becomes a classical phase space distribution function [13]. In terms of $\alpha$ (defined in section II B ) it is

$$\mathcal{W}(W(x,p)) \underset{\alpha \ll 1}{\simeq} \delta\left(W'(x,p) - W(x,p)\right) \underset{\hbar \ll 1}{\simeq} \delta\left(f' - f(x,p)\right) \qquad (7)$$

One can relax the $\alpha \ll 1$ assumption but not the $\hbar \ll 1$ assumption using Wigner functionals [49], defined over field configurations $f_{1,2}(x)$ in configuration space

$$W\left(f_1(x), f_2(x)\right) \simeq \int \mathcal{D}\phi(x) \exp\left[-if_2(x)\phi(x)\right] \left\langle f_1(x) + \frac{1}{2}\phi(x) \left| \hat{\rho} \right| f_1(x) - \frac{1}{2}\phi(x) \right\rangle \qquad (8)$$

where $\rho$ is the density matrix, defined via the partition function (See [50]) This expression is exact at the quantum level, and hence it's momentum equivalent is a straight-forward infinitely dimensional Fourier transform with $\tilde{f}_{1,2}(p)$. It also contains every possible correlation of the BBGKY hierarchy, encoded, in configuration space in "bunchings" between $f_1(x)$ and $f_2(x)$ as explained in the previous section ($f(x_1, x_2, ..., x_n)$ would be related to the $n - th$ cumulant of the functional).

Analogously how $f$ is the $\mathcal{O}\left(h^0\right)$ limit of $W$ [51] one can imagine the Boltzmann functional is the corresponding limit of the Wigner functional in [49], a decohered system with an

undefined probability density. More formally, the regime where the ansatze presented in this work are valid are in section 2 .

In this regime, where Eq. 7 is relaxed Eq. 6 would become something like this

$$\frac{p^\mu}{\Lambda}\frac{\partial}{\partial x^\mu}f_1(x,p) = \langle\mathcal{C}[f_1,f_2]\rangle_{f_2} - g\,\langle\mathcal{V}^\mu[f_1,f_2]\rangle_{f_2}\frac{\partial}{\partial p^\mu}f_1 \tag{9}$$

the left-hand side is identical, but the RHS is written in terms of

$$\langle O\rangle_{f_2} \equiv \int \mathcal{D}f_2 O(f_1,f_2)W(f_1,f_2) \tag{10}$$

with $\mathcal{C}, \mathcal{V}^\mu$ being the generalizations to functional averages of Vlasov and Boltzmann collision operators.

Physically, the we are "keeping track" of $f_1$ and letting $f_2$ represent our ignorance of the "real distribution". Hence, the Vlasov term can be physically interpreted as "an ensemble of forces defined by our ignorance of the real distribution" acting on a distribution $f_1$ of particles. The Boltzmann term is, analogously, an ensemble of collision terms.

Gaussianity in this context means that any phases between degrees of freedom oscillate "fast" w.r.t. any time-scale, so position and momentum decouple into a classical probability in both position and momentum space which is approximately of Gaussian form

$$W(f_1(x),f_2(x)) \simeq \rho[f(x,p),f'(x,p)] = \frac{1}{\mathcal{Z}}\exp\left[-\frac{D[f,f']}{2\sigma_f^2(x,p)}\right] \tag{11}$$

where $\sigma_f$ is some undetermined "width" function (whose significance will be clear shortly) and the obvious choice of a distance measure is

$$D[f,f'] = \int d^3x d^3p\,(f(x,p) - f'(x,p))^2 \tag{12}$$

with the Boltzmann-Vlasov equation recovered for $\sigma_f \to 0$. The large number theorem makes it likely that $\sigma_f \sim N_{DoF}^{-1/2}$, the square root of the number of degrees of freedom so it is certainly away from the ensemble average limit for the "small" fluids seen in hadronic collisions and ultra-cold atoms.

At first sight, the Gaussian approximation appears arbitrary and explicitly at odds with the aim to construct a theory "to all orders" in the BBGKY hyerarchy, since correlations $\mathcal{O}\left(f_1(x_1,p_1)\times f_2(x_2,p_2)\times f_3(x_3,p_3)\right)$ and higher could give rise to non-Gaussianities. The point is that Gaussianity in field theory has two quite distinct justifications: There is the

fact that free field theories, the basis of perturbative constructions, have Gaussian wavefunctions with free theory propagators (Eq. 13 with $G(x-y)$ being the Fourier transform of a free theory propagator). In this sense, higher order cumulants come associated with increasing powers of the coupling constant so Gaussianity is a good approximation to weak coupling.

However, in 3+1D Gaussian functional integrals are the only ones that are well-defined analytically. Not conincidentally, renormalization can be shown to arise from a central limit type expansion [37]. Examples such as superconductivity [38] variational QCD [39] and random matrix theory for nuclear physics [40] show that the Gaussian approximation, where the wave functional is of the form

$$|\Psi\left[\{\phi\}\right]\rangle \propto \exp\left[-\frac{1}{2}\int d^3x d^3y\,(\phi(x)-\phi_\infty(x))\,G^{-1}(x-y)\,(\phi(y)-\phi_\infty(y))\right] \quad (13)$$

where $\phi_\infty(x)$ is the topological term at infinity and $G^{-1}$ is the Green's function, is relevant for situations where coupling is strong. The renormalizeability connection to the central limit theorem [37] in fact makes this form close to unavoidable (bar some exotic examples discussed in [37]) , precisely because the well-definedness of a theory in the UV requires a convergent functional integral (thus, a "stable renormalized" $G(x-y), \phi$ emerging from a microscopic "bare" $G_{bare}(x-y), \phi_{bare}$ is nalogous to the approach to the central limit, and the limits on the number of couplings of $\phi_{bare}$ in the lagrangian is a consequence ). In this sense a visible breakdown of Gaussianity can be seen either as the appearence of irrelevant terms in the effective theory or as the breakdown of the classical probability approximation (which is the one "heuristic" requirement of the approach presented here). Higher order correlations will find their manifestation in the complexity and transendentality of the Green's function $G(x-y)$ (in BCS theory [38] a condensate emerges).

In the semi-classical (diagonal density matrix) limit Eq. 13 can only converge to something like Eq. 11 and Eq. 12 . Since for a renormalizeable theory irreducible diagrams are a maximum of $2 \to 2$) additional functional integrals of the type $\int \mathcal{D}f_2\mathcal{D}f_3 O(f_1, f_2, f_3)$ are also subleading in $\hbar$ and hence neglected in the semiclassical expansion. In terms of classical probability theory the "functional central limit" applies to the number of *events* from which $\{f\}$ is inferred. This is assumed here to be "large", even through each event could have "a small number" of degrees of freedom dominated by correlations of many particles. These are not included in higher cumulants *of the functional* (which would render it ill-defined) but

rather in the form of $\sigma_f(x,p)$ of Eq. 11 , just like multi-particle interactions and states are not precluded by a wavefunctional such as Eq. 13 , their probability is encoded in $G(x-y)$ (A good demonstration of this is the treatment of superconductivity [38]).

We note that a way to generalize the H-theorem is not apparent here, since microscopic entropy as it is usually defined will be inherently scale dependent [50]. However, a generalization of local equilibrium based around the vanishing of the RHS Eq. 9 is immediately apparent, as is apparent, from the corresponding "LHS=0" equation, the onset of ideal hydrodynamic behavior. The terms on the right hand side of Eq. 9 converge to

$$\mathcal{C}[f_1, f_2] = \int d^3 [k_{1,2,3}] \, \sigma_{scattering} \left( \; f_1(x,p) f_2(x,k_1) - f_2(x,k_2) f_1(x,k_3) \right) \tag{14}$$

where the Vlasov operator here $\mathcal{V}$ is the Vlasov operator

$$\mathcal{V}^\mu[f_1, f_2] = \int dx_{1,2} F^\mu(x_1 - x_2) \Theta((x_1 - x_2)^2) f_2 f_1 \tag{15}$$

Where $|M|^2$ is the scattering matrix element and $F^\mu$ the force field, augmented by a $\Theta$ function enforcing causality. By the analogy of Eq. 12 to the Green's function in Eq. 13 and the assumption of being close to the central limit theorem [37], we can conjecture that higher order terms in Eq. 15 and Eq. 14 will show up as corrections of $\sigma_f^2$ in Eq. 12 (representing oscillating condensates as such, as in [38]) to leading order in $\hbar$ if the classical ansatz is to stay well-defined.

Note that [22] one can consider the Boltzmann term as the UV completion of the Vlasov term, as scattering is the continuation "within the coarse grained cell" of the Vlasov evolution, increasingly unstable at smaller scales[2]. As "infinitely unstable infinitely local" interactions degenerate into random scattering, they are taken care of by the Boltzmann collision term, while long-range correlations are taken care of by the Vlasov drift term. Thus one expects, when one coarse grains, each term to be scale dependent but not the difference.

Very roughly, such "two independent scales up to a redundancy" parallel the Pauli-Villars renormalization scheme, where two infinities are needed to maintain Gauge symmetry [27]. Analogously, if Boltzmann is the "UV completion" of the Vlasov term (as was argued for

---

[2] [22] contrasts this instability in terms of KAM's theorem, which on the contrary implies the existence of an $\epsilon H_0 + H_I$, where $H_0, I$ are respectively integrable and non-integrable hamiltonians, where integrability is not broken. However this $\epsilon$ generally depends as $\mathcal{O}\left(e^{-N}\right)$ and to make the transition to a probability density function, $x_i, p_i \to f(x,p)$ requires $i \to \infty$, which nullifies the lower KAM limit. This is a heuristic explanation as to why Vlasov type equations are always unstable at all scales

in [22]), this can not result in physics depending on whether an interaction happens "via a Vlasov term above the cut-off" or a "Boltzmann term below it".

To see how this symmetry works physically we note that

- The integral in $\mathcal{C}$ is in momentum space while in $\mathcal{V}^\mu$ it is in position space. $f_{1,2}(x_{1,2}, p_{1,2})$ are of course defined in both but the gradients expected in any expansion will be,respectively,in position and momentum.

- For a consistent coarse-graing the scattering cross-section matrix elements and the long distance semi-classical potential are strictly related, as forces are related to scattering via potentials. For scalar particles

$$F^\mu(x) = \partial^\mu V(x) \quad , \quad \sigma_{scattering} \sim |M(k)|^2 d\Omega(k) \quad , \quad M(k) = \int d^3 x e^{ikx} V(x) \tag{16}$$

  with the appropriate extension for vector potentials. Thus in general both terms are present.

The point here is that for finite functional width $\sigma_f$ in equation 11, even away from the Gaussian parametrization, there arises a hidden "gauge" symmetry within the RHS of Eq. 6 . Consider all possible transformations such that

$$f(x,p) \to f'(x,p) \quad , \quad \underbrace{\left\langle \hat{C}\left(f(x,p), f'(x,p)\right)\right\rangle}_{lim_{f \to f'} \sim \partial f/\partial x} = \underbrace{\left\langle \hat{\mathcal{V}}^\mu\left(f(x,p), f'(x,p)\right)\right\rangle}_{lim_{f \to f'} \sim \partial f/\partial p} \frac{\partial f}{\partial p_\mu} \tag{17}$$

In the ensemble average Eq. 11 has no physical meaning because $f(x,p) \to f(x',p')$ can only be a shift in phase space, not a shift *in functions*.

In the Gibbsian picture, however, $f(x,p)$ is itself *unknown*, and only estimated via a coarse-graining. Hence the RHS of Eq. 9 can be dominated by redundancies so as to be qualitatively *very* different equation from Eq. 9 , even for small $\sigma_f$, ie narrow distributions in functional space. In other words, if only the Boltzmann term or the Vlasov term are present one assumes that as the Boltzmann functional converges to a $\delta$-functional,ie a function, the equation of motion for it converges to a Boltzmann or Vlasov equation. But if both terms are included, the redundancy in the difference spoils this convergence.

Physically, the manifestation of this is that the RHS of eq 6 vanishes in just two cases, free streaming and ideal hydrodynamics. This is because $\sigma_f \to 0$ and all dependence of it on

$x, p$ is irrelevant. However, for eq 9 there is a wealth of situations, parametrized by Eq. 17 where the system flow looks isentropic because $\sigma_f$ will be a complicated (perhaps fractal/non-integrable) function of $x, p$ which means many pairs $\{f(x, p), f'(x', p')\}$ exist whose difference is "small" with respect to it. In other words, there will be many configurations where the system will look like an ideal fluid, along the lines of [1]. This is shown schematically in Fig. 3 What happens is that close to the local equilibrium limit we do not know if the volume

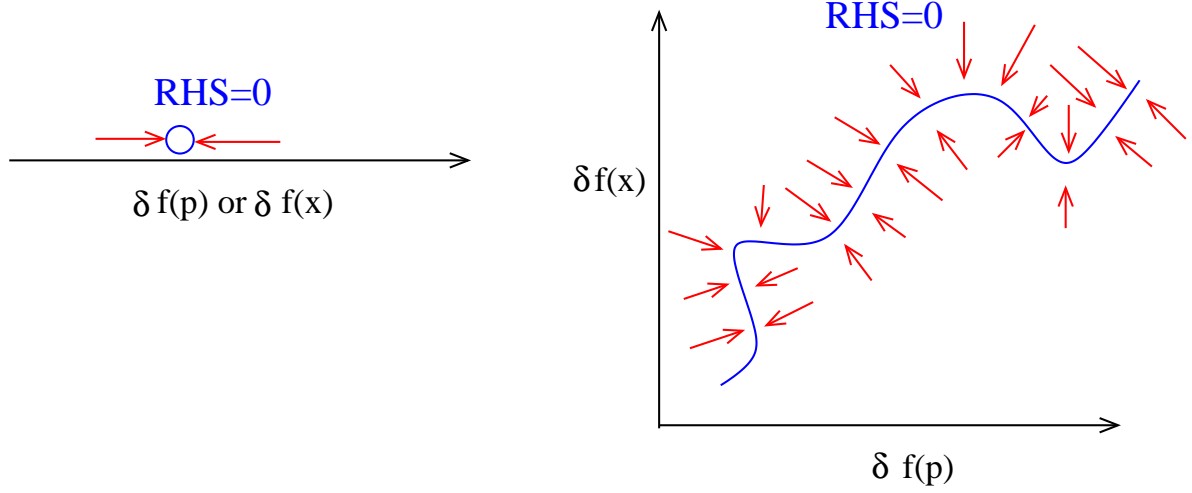

FIG. 3. A representation of how when both Boltzmann and Vlasov terms are considered the limit of the Boltzmann functional converging to a function admits a continuum of minima where Eq. 9 is indistinguishable from an ideal fluid dynamic equation. In this notation

$$\delta f(p) = \int dx \, (f(x, p) - f'(x, p)) \qquad , \qquad \delta f(x) = \int dp \, (f(x, p) - f'(x, p))$$

cell is being moved by *microscopic pressure* (described by a Boltzmann type equation) or rather by a *macroscopic force* (described by a Vlasov term). The set of configurations where a pressure gradient is exchanged for a force corresponds exactly to the set of configurations where the difference between the two sides of Eq. 9 does not change. In a Gibbsian picture, therefore, all such $f$ need to be counted in the entropy which generally results in differences w.r.t. the Boltzmannian entropy [26].

At the classical level, it has long been known [22] that the Boltzmann term can function as a "counter-term" cutting off the effect of short-range instabilities of the Vlasov term. At the quantum level this picture is indeed confirmed by the fact that the Boltzmann term describes "microscopic" and the Vlasov term "macroscopic" DoFs. The set of transformations leaving the RHS of eq. 9 invariant can be thought of as defining a "Gauge orbit" across the space

of $f(...)$ which are part of the same "Gibbsian" ensemble. Gibbsian here would mean that the observer has at their disposal an ensemble $\{x, p\}$, allowing them to infer a $\langle f(x, p) \rangle$ and it's uncertinity $\delta f(x, p)$, the latter related to a some scale defined by the coarse-graining in the distribution of $\{x, p\}$). Different functions in $\langle f(x, p) \rangle \pm \delta f(x, p)$ will have different Boltzmann and Vlasov terms but the same RHS of Eq. 9 . Note that the above argument needs to be updated with spin and vortical susceptibility if non-conservative fields, such as magnetic fields [23], are considered (the Gauge-like ambiguity would be if angular momentum is exchanged via vortical motion augmented with spin-orbit coupling or magnetic fields)

Quantitatively, these redundancies should ensure that the system is indistinguishable from local equilibrium in a much wider array of circumstances than a purely Boltzmann description would suggest. As there is no small parameter in the functional expansion around the average $f(x, p)$, an analytical quantification of this statement is non-trivial. However, the universality of random matrix theory could provide a quantitative validation of this point if causality is neglected.

## IV.   A NON-RELATIVISTIC INSIGHT FROM RANDOM MATRICES

Let us discretize the system (using $i$ for position and $j$ for momentum variables) and use random matrix theory, $f(x, p) \rightarrow f_{ij}(x_i, p_j)$ $\mathcal{C} \rightarrow \mathcal{C}_{i_1, i_2}, \mathcal{V} \rightarrow \mathcal{V}_{j_1, j_2}$. Of course we have neglected causality (the $\theta$-term in Eq. 15  as well as Lorentz invariance (broken in the discretization), but this is a round qualitative estimate, perhaps relevant for cold atom measurements such as [7, 8]

Equation 9 becomes of the form

$$\dot{f}_{ij} - \left[ \frac{\vec{p_k}}{\Lambda} . \Delta_k \right] f_{ij} = \int d\left[ f'_{i_1 j_1} \right] \left[ \mathcal{W}_{i_1 j_1 ij} \left( \mathcal{C}_{jj_1} \left( f_{ij} f'_{i_1 j_1} - f_{ij_1} f'_{i_1 j} \right) - \mathcal{V}^\mu_{ii_1} f_{ij} f'_{i_1 j_1} \frac{\Delta f_{ij}}{\Delta p^\mu} \right) \right] \quad (18)$$

$\Delta$ is the discrete derivative (the difference between lattice points normalized by lattice spacing) and $\mathcal{W}_{i, i_1, j, j_1}$ is a discretized version of Eq. 11  (in agreement with the definition Eq. 10 ). Double summation is used in the (...) bracket but $\mathcal{W}_{...}$ is multiplied separately. As we describe in detail below, the RHS can be thought of as a Gaussian random matrix ensemble

- $\mathcal{V}, \mathcal{C}$ are deterministic matrices of $i, j$. Hence, one can do a change of variables

$$d\left[f'_{i_1 j_1}\right] \to \left\{ \begin{array}{c} \mathcal{C}^{-1} \\ \mathcal{V}^{-1} \end{array} \right\} d\left[f'_{i_1, j_1}\right] \tag{19}$$

  This results in a series of Gaussian ensembles, with a transformed $\mathcal{W}$ as the weight, equivalent to a previous one up to a normalization factor. These ensembles are invariant under similarity transformations that mix $\mathcal{C}, \mathcal{V}$ with the distribution of $f_{ij}$, and in fact are related to the functional symmetries we are argue for.

- Provided the system is governed by central force type equations $\left\{ f_{i_1 j_2} f'_{i_2 j_2} \mathcal{V}_{i_1 i_2} \right\} \sim f\left(x - x'\right)^2 \hat{e}_{x-x'}$ (contracted with $\Delta f / \Delta p^\mu$, where $\hat{e}_n$ is th unit vector in direction $n$), is an antisymmetric ensemble in $i_{1,2}$. $j_{1,2}$ is traced over in a normalization factor

- $\mathcal{C}_{j_1 j_2} \left( f_{i_1 j_1} f'_{i_2 j_2} - f_{i_1 j_2} f'_{i_2 j_2} \right)$ is also an anti symmetric in $j_1 j_2$, $i_{1,2}$ are traced over in a normalization factor.

- This is however a deformed ensemble, since the average $\langle f_{ij} \rangle$ is non-zero.

- In the $\sigma_f \to 0$ limit of Eq. 11 one expects $\langle f_{ij} \rangle$ to reflect the general Boltzmann Vlasov estimate. More generally, we note $\mathcal{C}_{j_1 j_2}$ conserves momentum and $\mathcal{V}_{i_1 i_2}$ respects Lorentz invariance, so momentum conservation on average can be implemented via Lagrange multipliers where the quantity maximized is the local entropy given the choice of $d\Sigma_\mu$ (invariance w.r.t. $d\Sigma_\mu$ would then be enforced by the gauge symmetry described in the previous section and [1]). Thus one expects $\langle f_{ij} \rangle$ away from $\sigma_f \to 0$ will be of the form

$$\langle f_{ij} \rangle \propto \exp\left[-d\Sigma_\mu(x_i)\beta_\nu(x_i)p_j^\mu p_j^\nu\right] \tag{20}$$

  for some choice of $\beta_\mu, d\Sigma_\mu$ in line with the gauge-like expectations from [1]

This problem is the combination of ensembles studied for many decades [54, 55] but an elegant solution was shown in [56], where it was shown that the distribution is that of the Wigner semi-circle and outliers.

$$\rho(\lambda) = \rho_0(\lambda) + \frac{1}{N} \sum_{k, \lambda_k^* > J} \left(\delta(\lambda - \mu_k) + \delta(\lambda + \mu_k)\right) \quad , \quad \rho_0(\lambda) = \frac{1}{\pi J}\sqrt{1 - \frac{\lambda^2}{4J^2}} \tag{21}$$

the RHS of Eq. 9 will be the difference between two such "shifted" Gaussian ensembles.

$$\dot{f}(x, p, t) - \frac{\vec{p}}{m}.\nabla f(x, p, t) = N_p(t)F(J_p(f(x, p, t))) - N_x(t)\frac{\partial f}{\partial p}F(J_x(f(x, p, t))) \tag{22}$$

where $N_{p,x}$ are extra normalizations (from Eq. 19 ) and

$$F(J) = J \int \rho_0(x) e^{-x^2} dx + \sum \exp\left[-\mu_k^2[J]\right]$$

($J$ is perhaps related to the cut-off for the Vlasov and Boltzmann modes). Thus, assuming the "sparse" exponential terms can be neglected, the evolution will be driven by a difference between two Bessel-function type terms, where $N_{x,p}$ and $J_{x,p}$. This is much faster than the relaxation time of the Boltzmann equation, where the corresponding equation to Eq. 22 in the close-to-equilibrium (relaxation time) approximation is

$$\dot{f}(x,p,t) - \frac{\vec{p}}{m} . \nabla f(x,p,t) = \frac{f_0(x,p,t) - f(x,p,t)}{\tau_0} \tag{23}$$

The RHS for the "functional" term Eq. 22 is exponentially suppressed, while that of Eq. 23 is suppressed by $\mathcal{O}(f - f_0)$. The first is much more likely to stay close to zero even when the configuration of the system is far away from what would be called thermodynamic equilibrium.

Physically, a dynamics such as that of Eq. 22 means the system "rapidly and uniformly transitions to close to local equilibrium at the beginning" rather than approaching it slowly and inhomogeneusly. This is consistent with the Gaussian ansatz functioning throghout this dynamics, since the equilibrium state could be seen more as similar to a "condensate" (a' la' the dominant condensate in superconductivity and QCD [38, 39], appearing in Eq. 11 at the level of $\sigma_f$) than a complicated multi-body dynamical process that generally violates gaussianity.

Of course, the model presented here is highly acausal. Including causality in the Vlasov potential would add a non-trivial correlation to the random matrix which we do not at the moment know how to perform analytically.

### A.   Zubarev hydrodynamics and random matrices

The connection to random matrices of the above two sections can also be extended via Zubarev hydrodynamics. Consider a general strongly coupled system in a volume $V$ (Boltzmann-Vlasov, quantum chaos, whatever). Divide $V$ at a given time-step into a "random lattice" of a large $N$ points $\Sigma_\mu^{i=1...N}(t)$ such that

$$\Delta V \equiv \Delta^3 \Sigma^i \quad , \quad \sum_{i=1}^{N} \Delta V_i = \sum_{j=1}^{N} \prod_{j=1}^{3} \left(\Sigma_j^{i+1} - \Sigma_j^i\right) = V \tag{24}$$

Now take the output of the microscopic system over its evolution, and simply maximize the Zubarev statistical operator

$$\ln \mathcal{Z} = \ln \left[ \prod_{i=1}^{N} \exp \left( \Delta^3 \Sigma_\mu \left( \beta_\nu T^{\mu\nu} - \mu_i J^\mu \right) \right) \right] \tag{25}$$

with, as constraints, $\beta_\mu$ (for the energy-momentum current) and $\mu_i$ (for chemical potential). $d\Sigma_i$ are free parameters, under the constraint of total volume invariance. The resulting stochastic picture is very similar to that conjectured in [57].

This is a very complicated Lagrange multiplier problem (The number of multipliers is 3+1+Number of conserved charges per point), but it is still a linear problem. Since unstable solutions of "large systems of linear equations" approach random matrix theory ([40] and references therein) a connection is clear.

A numerical simulation is for now necessary to see how good is Zubarev hydrodynamics using a given highly non-linear system with many degrees of freedom. If $\ln \mathcal{Z}$ defined as Eq. 25 leads to a "large" value for the likelihood, some fluctuating hydrodynamics is a good approximation for the systems evolution. The dependence of this on the number of degrees of freedom is far from clear, in particular it is far from clear that it should always increase with the number of degrees of freedom. This was the main argument in [57], where it was conjectured that combining Zubarev hydrodynamics with Crooks fluctuation theorem is a good ansatz for relativistic hydrodynamics in the strong fluctuation regime, and in this regime, counter-intuitively, fluctuations "help" maintain the system in local equilibrium. Here, we see that this picture is related to the functional thermalization picture in the random matrix limit. A numerical simulation of the two would be able to confirm if this similarity is quantitative.

## B.    A numerical algorithm

The discretized Eigenvalue analysis in [52] (but also [53]) allows us to make an estimate of Eq. 9 initializing it close to a fluctuating equilibrium and seeing at every step if something like "an ideal hydrodynamic evolution" is maintained on average even at gradients of $\beta_\mu$ where a typical Boltzmann configuration would be far away from the hydrodynamic regime. The algorithm is summarized as follows

**(i):** Start with an average $\langle T_{\mu\nu}\rangle$. Create an ensemble $\{f\}$ of configurations in every cell $x_i, p_i$ using equation Eq. 25 for $P[f(x_i, p_i)]$, with a sample $\{T_{\mu\nu}\} \sim \{p_\mu p_\nu\}$ This can be done from a Metropolis type algorithm, with $\beta_\mu$ being Lagrange multipliers.

An immediate issue is the choice of $d\Sigma_\mu$, the foliation. Since we are simulating using a square grid around a hydrostatic limit [52], consistency requires $d\Sigma_\mu = dV\left(1, \vec{0}\right)$. One might need to check that the gauge-like symmetries with respect to a reparametrization of $\Sigma_\mu$ of [1, 24] will emerge. $\beta_\mu$ is given by the Landau condition $\beta_\mu T^{\mu\nu} \propto \beta^\nu$

**(ii):** Expand $\{f\}$ in Eigenvalues and Eigenvectors, according to to [52]

**(iii):** Evolve each $\{f\} \to \{f\}'$ according to the Boltzmann equation, using the Eigenvalue analysis of [52]. This time a Vlasov operator respecting causality can be constructed via Eq. 15

**(iv):** Construct $\langle T_{\mu\nu}\rangle\left(x_i\right) = \langle p_\mu p_\nu\rangle\left(x_i\right)$ and $\beta_\mu(x_i)$ from $\{f'\}$ and return to step (i)

If the picture argued for in this work is correct then, while perhaps the typical f in the ensemble at each step is far from the equilibrium value, fluctuations within the $\{f\}$ will smear out non-hydrodynamic effects and the evolution of $\beta_\mu(x_i)$ will follow the hydrodynamic description on average. Causality means this model would be different from the random matrix ansatz discussed in the previous section (the matrices would be "locally random" within a causal diamond) so a comparison would be interesting. Also, the "Gaussian width" $\sigma_f(x, p)$ of Eq. 12 can be numerically reconstructed step-by-step. The correctness of the picture suggested by this work would correspond to $\sigma_f(x, p)$ becoming more and more "rapidly oscillating" in $x, p$ in a way that makes outwardly different pairs of $f(x, p), f'(x, p)$ being equivalent for more intents and purposes.

## V. DISCUSSION

This has been a very speculative exercise. At the moment, we do not have a way to check quantitatively if a functional Boltzmann equation approach will give the desired result, an approach to local equilibrium which

- Is significantly faster than that of the Boltzmann equation

- Does not increase as the number of degrees of freedom goes down

At best, a "Galilean" model (instant signal propagation) can be looked at from a random matrix perspective, and universality of random matrix ensembles seems to show that indeed this scenario is plausible (Note that this universality has some similarity to the "inverse attractor" postulated in [1], where every system "looks" similar when sampled a certain way). Nevertheless, heuristically when the number of degrees of freedom is small ensemble average notions such as "phase space distribution function" are obviously inadequate and must be generalized, and a functional approach, with discretization, might be the best way to achieve this. Meanwhile,experimental tests of collectivity in smaller and smaller systems, both cold atoms [7, 8] and heavy ion collisions [3, 4] will tell us if a theoretical justification of hydrodynamics with small systems is worth pursuing while numerical implementations of the functional picture, described in the previous section, will show whether the rapid thermalization via the functional picture is indeed realized.

GT thanks CNPQ bolsa de produtividade 306152/2020-7, bolsa FAPESP 2021/01700-2 and participation in tematico FAPESP, 2017/05685-2. We thank Igor Gorniy and Leonid Pastur for providing references and answering my newbie questions about random matrices, Peter Arnold for explaining some subtleties of thermalization in quantum field theory and Leonardo Tinti, Sangyong Jeon and Stanislaw Mrowczynski for helpful discussions. The initial part of this work was done when I was in Kielce under NAWA grant BPN/ULM/2021/1/00039

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

DOI: 10.1007/s11511-011-0068-9

I recommend the presentations in

`https://www.youtube.com/watch?v=3Aa1Fyd1pW0`

and `https://youtube.com/watch?v=ZRPT1Hzze44`

For a great introduction on this topic from a mathematics perspective

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
