# Peer review of "The functional generalization of the Boltzmann-Vlasov equation and its Gauge-like symmetry"

_SciPost Physics_

## Round 1 · Referee Report · Anonymous (Referee 1) · 2023-11-21

Strengths

-The premise of the paper it interesting.

-It brings some alternative concepts in statistical mechanics that can be helpful in addressing the proposed problem

  • The author is honest about the degree of speculation

Weaknesses

-The paper is very speculative

  • The regime of validity of the ideas that the author proposes is fully relegated to the Appendix, without a sufficient discussion in the main text;

-Multiple typos, incomplete phrases and not well defined notation, which when accumulated with the previous points, impair the understanding of the paper;

Report

Dear Editorial Board and Author,

The present paper attempts to address the problem that hydrodynamics seems to work in systems with a small number of particles using a generalization of the Boltzmann equation, which considers the evolution of the single particle distribution function, that is now considered as a stochastic variable.

Among the expected acceptance criteria, I think this paper meets only the criterion of providing a link to new link between different research areas. Namely, some notions in non-equilibrium quantum field theory and hydrodynamics. It is indeed a possible way to regard the problem, but this not uncontroversial and to this point, extremely speculative.

I would not recommend the publication of this manuscript without major editing, since the accumulation of the various points below combined with the admitted high degree of speculation make it difficult to do otherwise. I apologize if the report is too long, but I think it is necessary, given that many unorthodox ideas are employed. I explain the points below. I hope they are clear enough.

1.1 Since the difference between the Gibbsian and the Boltzmannian notions of entropy is not usually discussed in this field and the papers [1,18] are by the same author, a few sentences about it could make the discussion more self-contained.

1.2 The current introduction has more than the context and the summary of achievements. It also has content that is part of the bulk of the paper. This contradicts general acceptance criteria of this publication.

  1. I did not understand the content of the phrase (second paragraph of the introduction): "Yet no indication exists that if we somehow tightly selected for initial geometry, we would not have extra uncertainties due to dynamics, the sign of 'perfect' hydrodynamics ". I don't see the link between the "the sign of 'perfect' hydrodynamics " and the remainder of the phrase.

  2. If I understand correctly, in Eq. (1) $\Lambda$ is a scale beyond which the equation would not be valid, but Eq. (4) is a completely different equation and yet it possesses the same parameter there. Are they really the same, or a different cut-off?

  3. On the last phrase of the paragraph below eq. (3), was $f(x_1, x_2,...,f_{n})$ really intended or a typo? If not a typo what does it mean? Are there momenta here? This is not defined neither in the introduction nor in the Appendix.

5.0 It would be interesting to discuss the difference between this approach and the Wigner function approach. Wouldn't the 'function' BBGKY hierarchy of ref. [13] encompass these fluctuations? Is the functional formalism a way to take into accounts higher moments in the 'function' BBGKY? 5.1 Subscript missing in Eq. (5)? $\langle O \rangle_{f_{2}}$ instead of $\langle O \rangle$ ? 5.2 Would not there be a minus sign in the exponent of Eq. (6)? 5.3 Would not $\sigma_{f} \sim N_{DoF}^{-1/2}$? 5.4 How does the Vlasov term reduces to a derivative in momentum space as required for the limit to the 'usual' Boltzmann-Vlasov equation is recovered? 5.5 Doesn't the collision term assume some BBGKY-like truncation ? Wouldn't the smallness of the system break even this "functional molecular chaos"? 5.6 $\vert M \vert^{2}$ is defined before it is even mentioned, in eq. 10. 5.7 The integration measures $dx_{1,2}$ and $d^{3}[k_{1,2,3}]$ are not defined

6.1 In Eq. (11), shoudn't $\hat{C}$ be $\mathcal{C}$? Why are there hats in $\mathcal{C}$ and $\mathcal{V}$? 6.2 The text below Eq. (11) has an incomplete sentence that impairs the understanding. "Away from a full ensemble average,"

7.1 In the last paragraph of p.6 in the phrase "(...) we do not know if the volume cell is being moved by microscopic pressure (...) $\bf{and}$ a macroscopic force ..." should not the highlighted 'and' be an 'or' in a 'either' ... 'or' sentence?

7.2 In the last phrase of p. 6, does the author mean $\mathcal{V}[f]$ ($\mathcal{V}^{\mu}[f_1,f_2]$ ?) or $\langle \mathcal{V}^{\mu}[f_1,f_2] \rangle$ ? I would expect that the average should have some redundancy, not the operator itself.

7.3 In Fig. 1, are $\delta f(x)$ and $\delta f(p)$ marginal distributions of deviations from local equilibrium (e.g. $\delta f(p)$ is $\delta f(x,p)$ integrated over x)? Please, emphasize.

8.1 What is the definition of $\Delta_{i}^{\mu}$ and integration measure, $d[f'{i]$} j_{1}, in Eq. 12 ? 8.2 Is not there a missing index in $\partial f/\partial p$ in Eq. 12? Wouldn't the specific index change the discussion that follows? 8.3 Usually in random matrix theory, it is assumed that the ensemble is invariant under similarity transformations $M \mapsto M' = U^{-1}M U$, where U is unitary, and Eq. 13 reminds me of that. How do we see that $\mathcal{C}$ and $\mathcal{V}$ have the correct properties so that transformation (13) is valid? 8.4 Since the phase space has been discretized, is not Lorentz invariance also discretized? If yes, it is worth emphasizing it. 8.5 Is ${f_{i_1 j_{2}} f'{i_2 j}} \mathcal{V{i_1 i$}} } \propto \langle x - x' \rangle^{-2 an assumption or a result from random matrix theory? Please, this should be made clear. 8.6 What is being extremized so that Lagrange multipliers are considered in Eq. (14)? Is it functional (6) or some entropy functional, that is not defined? 8.7 The non-linearity of Eq. 12 should lead to the rising of multiplicative noise, right? Is one of the assumptions that these effects are small, even classically?

9.1 In Ref. 33 $\rho(\lambda)$is the eigenvalue density function, what would be its relation with $f_{ij}$ in the present case, the probability density of eigenvalues of $f_{ij}$? 9.2 Since $J$ is related to the maximum (and the minimum) eigenvalues in the continuum part of the distribution, I would expect $J_p$ and $J_x$ in Eq. 16 be related to the cut-off $\Lambda$, why it is instead $\sim \langle x \rangle $ and $\sim \langle p \rangle $?

  1. How can one see that the highly non-linear combination of $N_{x,p}$ and $J_{x,p}$ are small (since the author claims that the RHS of 16(?) is negligible) and lead to a 'relaxation time' much smaller than the relaxation time of the Boltzmann equation, which would lead collision term to grow? Is there a compensation between the collision and the Vlasov terms?

  2. Should there not be indices $j$ in the momenta in Eq. 17?

  3. Would the ensemble on step (i) in section III be created with a Metropolis-like algorithm?

  4. Is the universality evoked in the second paragraph of the discussion section related to an attractor-like universality, in which the system 'forgets' the non-hydrodynamic modes and an free-streaming, 'asymptotically ideal' hydro emerges?

  5. In sec. 2 of the appendix, is the BBGKY hierarchy referred the common BBGKY or the functional one? Could you elaborate on what is "non-perturbative" in this context?

  6. The common semi-classical expansion of quantum kinetic theory is motivated by the smallness of the wave packet? What is the suggestion of the present functional semi-classical expansion? The locality of the wave-functional packet? How is that different?

Best regards,

Anonymous Referee

Requested changes

Dear editors and author,

Below I describe the changes I would request from the author:

A. Address item 1.1 in the report;

B. Address item 1.2 in the report;

B.1 I recommend that the author starts a new section after the paragraph
ending in "and placing it within the more conventional
transport theory have been left to the appendix", before equation (1).

B.2 The beginning of such new section the author should discuss, at least
briefly, in one paragraph, the regime of validity of the assumptions. I think
this cannot be fully relegated to the appendices. A summary of section 2 of
the appendix should be enough.

C. Address possibly incomplete/ambiguous phrases in items 2, 6.2, 7.1, 7.2, 7.3 of the Report;

D. Address possible typos in items 3,4, 5.1, 5.2, 5.3, 6.1, of the report;

E. Address the notation problems pointed in items 5.6, 5.7,8.1,8.2,11 of the report;

F. Address questions in items 5.0,5.5, 8.3, 8.4, 8.5, 8.6, 8.7, 9.1, 9.2, 9.3, 10, 11, 12, 13, 14, 15 of the report.

Best regards,

Anonymous Referee

---

## Round 1 · Referee Report · Anonymous (Referee 2) · 2023-12-2

# Report: arXiv:2309.05154

## I.  WHAT HAS BEEN STUDIED IN THIS PAPER ?

**This is a very interesting work where the physics of small systems and their hydrodynamic behaviour has been discussed in the context of Boltzmann-Vlasov equation.**

In this work, the author has discussed the onset of hydrodynamic behaviour of small systems by attributing such a phenomena to fluctuations in the system that enter in to the distribution function $f(x, p)$. In small systems, the distribution function is not known and should be guessed. According to this work, the change in the distribution function can be modelled as a *Gibbsian*, through Bayesian analysis. The starting point of this study is the Boltzmann equation, in which an extra term, known as the Vlasov term has been introduced, thus promoting the equation to Boltzmann-Vlasov equation. The features / purpose of this Vlasov term is the following :

- The introduction of the Vlasov term $(\mathcal{V}^\mu)$ is justified because for *small-systems which are highly correlated*, the instabilities in the system require a term which is of $\mathcal{V}^\mu$ type.

- The long-range interactions are taken care of through the $\mathcal{V}^\mu$ term in the Boltzmann-Vlasov equation. The collision term $(C[f])$ in the Boltzmann equation takes care of the quantities calculated from the scattering elements $(|\mathcal{M}|^2)$ viz. cross sections, decay widths etc.

The distribution function has been represented as a functional by expressing it as a probability density in the form of a Gaussian ansatz. This has been obtained via technique of *Wigner functions*. The assumed Gaussian ansatz for the probability density of distribution function, carries a functional width $\sigma_f \sim \sqrt{N_{\mathrm{DoF}}}$, which dictates how the probability density for the distribution function varies with the degrees of freedom of the system. A gauge redundancy has been claimed to exist even in the limit of narrow $\sigma_f$ and is responsible for the fluidic behaviour of small systems.

## II.  QUESTIONS / COMMENTS

1. In Eq. (2), the assumption of the distribution function is not very clear. A sketch of some calculations showing how Eq. (2) is derived, would be helpful.

2. It has been argued that the introduction of the *Vlasov term* and further using a Gibbsian estimation of the distribution function, induces a *gauge redundancy* in the system.
   Does the Vlasov term act like a background gauge field in the system ? A comparision made with the Boltzmann-Equation in the presence of background magnetic field shows that a force term coming from magnetic field looks similar to the $\mathcal{V}^\mu$ term in this paper.
   It is requested to look into :

   (1) Ref - G.S. Denicol *et.al* [1]

   (2) Ref - A. Dash *et.al* [2]

3. In Pg - 7, the following claim has been made : *" The set of transformations leaving the RHS of eq. 4 invariant can be thought of as defining a "Wilsonian" flow across the space of f (...) which are part of the same Gibbsian ensemble."*. Some explanation is required, particularly in the UV and IR regimes that how the RHS of Eq. (4) remains invariant under such transformations. Is the set of transformations that leaves Eq. (4) invariant related to the *scale transformations and local scale transformations* encountered in RG flow ? It would be useful if some explanation can be offered in this regard.

4. It has been argued in Sec. 2 of Appendix, that for a *strongly correlated system* the BBGKY hierarchy cannot be used as an expansion, rather a non-perturbative technique be used for this purpose. But a non-perturbative method such as *Mean Field Theory* might be used for this purpose -

   (1) Ref - M.B. Pinto [3]

   (2) Ref - P. Romatschke [4]

   This might be looked into for studying a strongly coupled system and some explanations are required for this in the manuscript. Please include these references in the manuscript which may offer scope for interesting studies in the future.

5. There are some typos in the manuscript which need to be corrected.

[1] G. S. Denicol, E. Molnár, H. Niemi and D. H. Rischke, "Resistive dissipative magnetohydrodynamics from the Boltzmann-Vlasov equation," Phys. Rev. D **99**, no.5, 056017 (2019) [arXiv:1902.01699 [nucl-th]].

[2] A. Dash, S. Samanta, J. Dey, U. Gangopadhyaya, S. Ghosh and V. Roy, "Anisotropic transport properties of a hadron resonance gas in a magnetic field," Phys. Rev. D **102**, no.1, 016016 (2020) [arXiv:2002.08781 [nucl-th]].

[3] M. B. Pinto, "Three dimensional Yukawa models and CFTs at strong and weak couplings," Phys. Rev. D **102**, no.6, 065005 (2020) [arXiv:2007.03784 [hep-th]].

[4] P. Romatschke, "Quantum Field Theory in Large N Wonderland: Three Lectures," [arXiv:2310.00048 [hep-th]].

---

## Editorial Decision

resubmitted